# Effect of Chitosan on Ruminal Fermentation and Microbial Communities, Methane Emissions, and Productive Performance of Dairy Cattle

**DOI:** 10.3390/ani13182861

**Published:** 2023-09-08

**Authors:** Jagoba Rey, Xabier Díaz de Otálora, Raquel Atxaerandio, Nerea Mandaluniz, Aser García-Rodríguez, Oscar González-Recio, Adrián López-García, Roberto Ruiz, Idoia Goiri

**Affiliations:** 1NEIKER—Basque Institute for Agricultural Research and Development, Basque Research and Technology Alliance (BRTA), Department of Animal Production, Campus Agroalimentario de Arkaute s/n, 01192 Arcaute, Spainxdiaz@atb-potsdam.de (X.D.d.O.); ratxaerandio@neiker.eus (R.A.); nmandaluniz@neiker.eus (N.M.); aserg@neiker.eus (A.G.-R.); rruiz@neiker.eus (R.R.); 2Leibniz Institute for Agricultural Engineering and Bioeconomy (ATB), Department of Technology Assessment and Substance Cycles, Max-Eyth-Allee 100, 14469 Postdam, Germany; 3Departamento de Mejora Genética Animal, INIA-CSIC, Ctra. La Coruña km 7.5, 28040 Madrid, Spain; gonzalez.oscar@inia.csic.es (O.G.-R.); adrian.lopezgar@upm.es (A.L.-G.)

**Keywords:** chitosan, fermentation efficiency, methane, whole-genome sequencing

## Abstract

**Simple Summary:**

Expanding our understanding of the changes induced by chitosan supplementation on the rumen microbiome and its complexity is not only a key factor in finding strategies to decrease CH_4_ production from ruminants but also in enhancing our knowledge on the mechanisms of action of chitosan in the rumen. Therefore, this study aimed to expand the knowledge about the activity and mode of action of chitosan on methanogenesis and rumen microbial taxonomy. This is the first study, to our knowledge, accounting for the effects of chitosan on ruminal microorganisms on in vivo conditions and using whole-metagenome sequencing. Chitosan did not affect rumen microbial diversity but induced shifts in the relative abundance of some microbial taxa. Chitosan increased relative abundance of Anaeroplasma and tended to reduce fibrolytic fungi and protozoa, resulting in a shift of ruminal fermentation towards a lower acetic to propionic ratio. However, CH4 emissions, microbial protein synthesis, and productive performance were not affected by CHI supplementation. The dose of chitosan used in this study seemed not to be large enough to have a concomitant improvement in animal performance and CH4 reduction. Therefore, future works with higher doses would be necessary to assess the potential use of this additive as methane inhibitor.

**Abstract:**

This study aimed to expand the knowledge about the activity and mode of action of CHI on methanogenesis and rumen microbial populations in vivo. A total of 16 lactating dairy cows were distributed in two groups, one of them receiving 135 mg CHI/kg body weight daily. The effect on productive performance, milk composition, fermentation efficiency, methane emissions, microbial protein synthesis, and ruminal microbial communities was determined. Supplementation with CHI did not affect rumen microbial diversity but increased the relative abundance (RA) of the bacteria *Anaeroplasma* and decreased those of rumen ciliates and protozoa resulting in a shift towards a lower acetic to propionic ratio. However, no effect on milk yield or methane intensity was observed. In conclusion, supplementing 135 mg CHI/kg body weight increased the RA of *Anaeroplasma* and decreased those of rumen ciliates and protozoa, both being related to fiber degradation in the rumen in different ways and resulted in a shift of ruminal fermentation towards more propionate proportions, without affecting CH_4_ emissions, milk yield, or milk composition. Further research with higher doses would be necessary to assess the potential use of this additive as a methane inhibitor.

## 1. Introduction

Ruminants consume cellulosic non-human edible food sources and, in turn, produce milk and meat, sources of high nutritional quality proteins. In the process of using and taking advantage of these cellulosic feed resources, ruminants produce greenhouse gases (GHG), primarily in the form of enteric methane (CH_4_), which is released to the environment through exhalation and eructation [1]. Methane is a GHG with a warming potential 27 times greater than that of carbon dioxide [2], thus contributing significantly to the carbon footprint of dairy farming. It has been reported that the livestock sector contributes 14.5% of all anthropogenic emissions, of which 3.3% are associated with the enteric CH_4_ [3]. At the same time, enteric CH_4_ is one of the pathways of energy loss in ruminants, which can reach 12% of the gross energy intake in lactating cows [4]. Therefore, CH_4_ emissions are not only a global warming issue but also a productive one. In this context, during the last years, reducing livestock GHG emissions and enteric CH_4_ is posed as a one of the main concerns in the agribusiness sector [5].

The formation of CH_4_ in the rumen is accomplished by a degradation and fermentation of the feed by bacteria, protozoa, and fungi. As a consequence of fermentation, volatile fatty acids (VFA) are formed, providing energy to the animal while releasing hydrogen (H_2_). Although there are several possible substrates for methanogenesis in the rumen that derive from bacterial fermentation (i.e., formate, acetic, and methyl compounds), H_2_ and carbon dioxide (CO_2_) are the most prominent ones [6,7]. Thus, there is a strong connection between the microbial fermentation processes, their H_2_ production, and CH_4_ formation by the methanogenic archaea in the rumen [3].

Strategies for CH_4_ mitigation through microbial community manipulation are varied and include, among others, vaccines against methanogens [8], probiotics [9], defaunation of protozoa or methanogens [10,11,12], and use of natural compounds present in plants [13,14,15,16]. Although there is extensive knowledge on the effects of these methods on microbial populations and CH_4_ emissions, further research regarding the identification and control of the interactions between different microbes in the rumen (i.e., Archaea, Prokaryote, and Eukaryote), which will result in optimum rumen fermentation, is needed. In this sense, classifying the microbial community based on a preserved gene is presented as a challenge associated with simultaneously analyzing different superkingdoms, as often the resolution is compromised, especially when the species are closely related. This method is also inherently limited because of the bias induced by the Polymerase Chain Reaction while amplifying the gene. In contrast, whole-metagenome shotgun analyses are accomplished by unrestricted sequencing of the genomes of most microorganisms present in a sample, including currently uncultured organisms. In this sense, metagenomic analysis of entire rumen microbial communities provide new perspectives on how methanogens interact with other members of this ecosystem and how these relationships may be altered to reduce methanogenesis [17].

Chitosan (an N-acetyl-d-glucosamine polymer, CHI) is a natural, nontoxic, biodegradable biopolymer [18] derived from deacetylation of chitin, a major component of the exoskeleton of crustaceans and insects. The antimicrobial mechanism of CHI is complex and has been related to cell surface and outer membrane damage and leakage of intracellular substances, leading to the death of microorganisms [19]. The antimicrobial activity of CHI has emerged as one of its most interesting properties [20], which led to evaluation of its use in ruminant nutrition [21]. Benefits observed in vivo trials seem to be caused by changes in ruminal fermentation, in particular, by increased propionic acid proportion and decreased acetic to propionic ratio [22,23,24,25]. Moreover, the theoretical decrease in metabolic hydrogen production [26] could lead to energetically more efficient fermentation patterns, which, in some works, have led to an improved feed efficiency in dairy ewes [27] and cows [28,29] due to the addition of CHI. However, the effect of CHI on enteric CH_4_ emissions has not been widely studied, and, sometimes, results have been contradictory. In this sense, while in vitro studies [26,30] have observed a substantial reduction in CH_4_ emissions (42 to 43%), Henry et al. [31] reported that CHI had no effect on enteric CH_4_ production in beef cattle. In vitro studies are often used to screen for the effect of an additive on ruminal fermentation, but, although being adequate for screening purposes, they have also some limitations, and, often, more important effects was observed in vitro compared to the whole animal [32]. Very few works have studied the effect of CHI on CH_4_ emissions under in vivo conditions, and, also, to our knowledge, none were performed in dairy cows. Therefore, more in vivo studies are necessary to identify the effect of CHI on methane yield and intensity under field conditions, mainly in dairy cows.

As previously stated, it is possible that changes in the rumen’s microbial structure can also affect CH_4_ production since it is mainly responsible for determining the fermentation routes. However, there are few works in the literature that studied how the inclusion of CHI in a diet affected microbial populations. A simplification of the structure of the bacterial community and a decreased abundance of cellulolytic bacteria under in vitro conditions was reported with an increased abundance of amylolytic bacteria when CHI was supplemented, which could explain the results observed on the fermentation routes [26,33]. Zanferari et al. [29] reported a reduction in bacterial species’ abundance, such as the Butyrivibrio group and *B. proteoclasticus*, related to the rumen biohydrogenation of fatty acids in dairy cows supplemented with CHI. However, to our knowledge, there are no works in the literature performed under in vivo conditions using whole metagenome sequencing studying the effect of CHI on ruminal microbiome and methane emissions. Therefore, a comprehensive characterization of microbial community richness and structure is essential under in vivo conditions with simultaneously analyzing different microbial superkingdoms to confirm the mechanism underlying the reported impact of CHI on rumen fermentation and CH_4_ emissions.

We hypothesized that CHI would affect ruminal microbial populations on dairy cows, leading to a shift in the ruminal fermentation with a reduction in methane production. Therefore, the aim of the present study is to expand the knowledge about the effects of CHI on ruminal, entire-microbial populations in in vivo conditions in lactating dairy cows and its effect on methanogenesis and production performance.

## 2. Materials and Methods

### 2.1. Animals, Experimental Design, and Diets

Cows were kept at the Fraisoro Farm School (Zizurkil, Spain) in loose housing conditions. All animals were in good health conditions and were monitored by qualified personnel during the study for health problems. A total of 4 lactating Holstein Friesian and 12 Brown Swiss dairy cows were paired based on breed, parity, days in milk (DIM), and milk yield during a 2-wk covariate period. Average DIM, body weight (BW), and milk yield of the cows before the beginning of the trial were (mean ± SD) 246 ± 129 d, 634 ± 64.7 kg, and 25.6 ± 4.58 kg/d, respectively. All cows were individually fed the control concentrate (CTR) during the covariate period, and then cows within a pair were randomly assigned to the control concentrate only, the control concentrate supplemented with 135 mg CHI/kg BW per day, or control group, without supplementation. Chitosan (Chitoclear. Deacetylation degree: >95%; viscosity: <500 mPa s; Trades S.A., Barcelona, Spain) was individually supplemented once a day as a powder and hand-mixed with a portion (5 kg on as fed basis) of the daily allowance of concentrate. Chitosan dose was chosen based on previous works with the same chitosan carried out with sheep [22]. Concentrate was formulated to satisfy production needs based on INRAE recommendations considering milk production and DIM. All cows had free access to a maize- and grass-silage diet offered once a day in the morning. The ingredients and chemical composition of the concentrate and basal forage diet can be seen in Appendix A. After the covariate period, the first two weeks were for adaptation to the diets, and measurements were taken during the following 5 weeks.

Cows were milked with an automatic milking system (AMS, DeLaval, 2004, Tumba, Sweden). Part of the concentrate was offered at the time of milking, and the other part (5 kg) was offered individually in buckets. The quantity offered in the AMS was established according to the maximum concentrate allowance per animal. All cows had free access to the AMS 22.5 h/d (a total of 1.5 h was dedicated to the cleaning of the system). Cows were granted milking permission after 6 h from previous milking, unless a milking failure occurred, in which case cows would be granted permission to be milked again immediately. In general, when the time elapsed since last milking was more than 12 h during the day, the cow would be fetched and forced to visit the AMS.

### 2.2. Measurements and Samplings

Daily individual milk yield (MY) was recorded at each milking by the AMS. Milk samples were collected from the AMS at each milking on the last day of the covariate period and once a week thereafter and stored with azidiol (3.3 mL/L) at 4 °C for fat, protein, and lactose determinations (Instituto Lactológico de Lekunberri, Lekunberri, Spain). Offered maize- and grass-silage and concentrates were sampled on a weekly basis to characterize their chemical compositions.

Daily individual CH_4_ concentrations were measured throughout the experimental period. Methane measurement was made by means of a Non-Dispersive Infrared Detector device (NDIR, Guardian NG Edinburgh Instruments Ltd., Livinstong, UK, 0 to 10,000 ppm range), as described in López-Paredes et al. [34]. An NDIR device sampling tube was placed in the feed bin of the AMS, and methane readings were recorded in each visit of the cows to the AMS. Briefly, air was sampled continuously at a rate of 1 L/min through the polyamide sampling tube (8 mm in diameter and 4 m long) from the front of a cow´s head to the gas analyzer to continuously measure CH_4_ concentration in the cow´s breath. Methane concentration was recorded at 1 s intervals and stored in a datalogger (Data Recorder SRD-99; Simex Sp. z o.o, Gdańsk, Poland). The NDIR device was verified during the installation, using standard mixtures of CH_4_ in nitrogen (0.0%, 0.25%, 0.5%, 0.75%, 1.0%; MESA International Technologies INC, Santa Ana, CA, USA). All measurements were corrected by a dilution factor to avoid the loss of concentration of methane dissipated in the air from the moment it was exhaled until it reached the tube. To calculate the dilution factor, a balloon was filled with a known methane concentration (CH_4_ 1.0%), then the air contained in the balloon was released at the inlet of the sampling tube placed in the AMS feeder, and the methane concentration subsequently recorded [35]. This process was repeated several times to obtain a mean dilution factor.

The animals’ BW was determined on days 1 and 7 of the covariate period. Both measurements were made using an automated weighing scale (ICONIX FX1 range 0–2000 kg; Ramaderia Casanova, S.L., Barcelona, Spain).

In week 3 of the sampling period, rumen samples were collected over two consecutive days for VFA determinations and DNA extraction for metagenomic studies. Sampling was at 00:00 and 12:00 h on d 1 and 06:00 and 18:00 h on d 2. Ruminal samples were collected from each dairy cow using a stomach tube (18 mm diameter and 160 cm long) connected to a mechanical pumping unit (Vacuubrand ME 2SI, Wertheim, Germany). The ruminal content was filtered through four layers of sterile cheesecloth. To analyze DNA, a sample of 50 mL of each ruminal extraction was placed immediately into a container with liquid N_2_ until it was stored in the laboratory at −80 °C. In addition, 15 mL of each ruminal extraction were separated into individual tubes for VFA analysis. Samples were immediately placed in the fridge at 4 °C until they were stored frozen at −20 °C ± 5 °C in the laboratory for further analysis.

In week 5 of the sampling period, during 4 consecutive days, individual spot urine samples (approximately 300 mL each) were collected from each cow at 12:00 and 00:00 (d 1 of each sampling period), 9:00 and 21:00 (d 2), 06:00 and 18:00 h (d 3), and 03:00 and 15:00 (d 4). Urine samples were collected by massaging the vulva. Urine was acidified (pH < 3) using 2M H_2_SO_4_ and were kept frozen until analyzed for urine derivatives. Blood samples (10 mL) were collected also in week 5, 1 h after morning feeding via coccygeal venipuncture into plain evacuated tubes without anticoagulants and in EDTA tubes (Venoject ^®^, Albet Comercial, Barcelona, Spain), for serum and plasma analyses, respectively.

### 2.3. Handling and Laboratory Procedures

#### 2.3.1. Feed

Maize- and grass-silage and concentrate were dried in a forced-air oven (48 h, 60 °C) and ground to pass a 1 mm sieve. The DM content (method 934.01) was then measured following the Association of Official Analytical Chemists [36]. Nitrogen concentration was determined using a Kjeltec Auto 1030 (Foss, Hillerød, Denmark) based on the macro-Kjeldahl procedure. The neutral detergent fiber concentration (NDF) was determined using an alpha amylase, but without sodium sulfite, and was expressed as ash-free [37]. Acid detergent fiber (ADF) was determined and expressed, excluding residual ash [38]. The fat concentration was analyzed without hydrolysis through the automated Soxhlet method (Selecta S.A., Barcelona, Spain) using hexane for 6 h as solvent. Starch concentration was determined by polarimetry [39].

#### 2.3.2. Milk

Milk fat, protein, and lactose concentrations were analyzed by near-infrared spectroscopy (Foss System 4000, Foss Electric, Hillerød, Denmark; Instituto Lactologico Lekunberri, Lekunberri, Spain).

#### 2.3.3. Volatile Fatty Acid Determinations

The analysis of VFA (acetic, propionic, butyric, isobutyric, valeric, and isovaleric) of the samples of ruminal fluid was performed by gas chromatography using a flame ionization detector. A volume of 4 mL of diluted ruminal fluid mixed with 1 mL of a solution of 20 g/L of 4-methylvaleric acid as an internal standard, in 0.5 N HCl, was centrifuged (15,000× *g* for 15 min at 4 °C) to separate the liquid phase from the feed residuals. After, the liquid phase was microfiltered (premium syringe filter regenerated cellulose, 0.45 µm, 4 mm, Agilent Technologies, Madrid, Spain), and 0.5 µL of liquid phase was directly injected in the chromatograph (Agilent 6890 N) using a capillary column (30 m × 530 um; 1-µm particle size; HP- FFAP, Agilent, Spain) and kept at 300 °C in the injector with a hydrogen flow 40 mL/min, air flow 400 mL/min, and make up (nitrogen) 25 mL/min flow. The injection loop was 20 µL. The individual VFA were identified using a standard solution of 4.50 mg/mL of acetic acid, 5.76 mg/mL of propionic acid, 7.02 mg/mL of butyric acid and isobutyric acid, 8.28 mg/mL of valeric acid, and isovaleric acid in 0.1N H_2_SO_4_ (A6283, P1386, B103500, I1754, 240370, 129542, respectively; Sigma-Aldrich, Madrid, Spain). The quantification was performed using an external calibration curve based on the standards described above. Data were expressed in mmol/100 mmol.

#### 2.3.4. Blood

Blood samples preserved in plain evacuated tubes without anticoagulants were incubated at room temperature and then centrifuged (1500× *g* for 15 min at room temperature). Then, serum samples were analyzed using an autoanalyzer (Saturno 150, CronyInstruments, Italy) for concentrations of blood urea nitrogen (BUN, Spinreact, Girona, Spain). Insulin-Like Growth Factor-1 (IGF-1) was analyzed by chemiluminescence (Maglumi 800, Snibe Diagnostics, Shenzhen, China) with the associated commercial kit (Maglumi CLIA, Snibe Diagnostics, Shenzhen, China). Blood samples preserved in EDTA tubes were centrifuged (1500× *g* for 15 min at 4 °C), and plasma was analyzed for glucose (glucose oxidase/peroxidase method [40]).

#### 2.3.5. Purine Derivative Determinations

Composited urine by animals were centrifuged, diluted (1:5), filtered (0.22 µm Millipore filter), and analyzed for purine derivatives (PD) by high performance liquid chromatography (HPLC), using a Shimadzu HPLC system equipped with a UV detector (205 µm) and two C18 reserved-phase columns (250 × 4.60 mm) connected in series with the mobile phase NH_4_H_2_PO_4_-acetonitrile (80:20) gradient at variable flow rate between 1.0–1.4 mL/min according to the method of [41], using a 0.03 M KH_2_PO_4_ buffer solution and using allopurinol as an internal standard for the quantification. The peaks were identified by comparing their retention times with those of known standards [42].

#### 2.3.6. DNA Extraction and Sequencing of Rumen Samples

Ruminal samples were thawed overnight (4 °C) and then homogenized in a blender. For each cow, a pool was made with 4 time-spot samples of ruminal liquid to perform DNA extraction. Finally, DNA of a homogenized sample of 250 µL was extracted using the commercial “DNeasy Power Soil” kit (QIAGEN, Valencia, CA, USA). The genomic DNA concentrations and their purity were measured using a Nanodrop ND-1000 UV/Vis spectrophotometer (Nanodrop Technologies Inc., Wilmington, DE, USA) with ratios 260/280 and 260/230 around 1.8 and 2.0, respectively. One µg of DNA from each sample was used as initial material for sequencing, following the ligation sequencing kit (SQK-LSK109) protocol from Oxford Nanopore Technologies (ONT, Oxford, UK), in a MinION (Mk1C) sequencer (ONT, Oxford, UK). Samples were multiplexed up to 8 samples in each run with the 1D Native barcoding genomic DNA (EXP-NBD104) ONT kit. The barcoded samples (700 ng of DNA in total) were pooled in a 1.5 mL Eppendorf DNA LoBind tube to perform adapter ligation for sequencing using a R9.4.1 flow cell.

#### 2.3.7. Bioinformatics

Base calling was made using Guppy toolkit (ONT; HAC option). Quality control was performed with FASTQC software (http://www.bioinformatics.babraham.ac.uk/projects/fastqc/ accessed on 7 June 2023), removing sequences with QS < 7 and length < 150 bp, and trimming by Trim Galore (https://www.bioinformatics.babraham.ac.uk/projects/trim_galore/ accessed on 7 June 2023).

Sequence analysis was performed using SqueezeMeta (SQM) pipeline for long reads [43], which performed Diamond Blastx against GenBank nr taxonomic database, then identifying and annotating open reading frames using the LCA method for taxonomy (based on e-value and identity scores). This tool was specifically developed to process long reads from ONT.

All sequences mapped as non-microbial (i.e., viruses, animals, and plants) were discarded. Microbial sequences were then filtered by prevalence (we discarded those genera present in less than 4 animals) to reduce data sparsity and sequencing errors and, by abundance, discarding those genera with a relative abundance lower than 0.0005%.

Alpha diversity was calculated using phyloseq [44].

Considering the compositional nature of metagenomic data, a CLR method [45] was applied using the unweighted option of the CLR function from the easyCODA R package [46] as follows:$$X_{CLR}=\left[\log\left(X_{1}/G\left(x\right)\right),\log\left(X_{2}/G\left(x\right)\right)\ldots \log\left(X_{D}/G\left(x\right)\right)\;\right]$$
with $G\left(x\right)=\sqrt[{D}]{X_{1}*X_{2}*\ldots *X_{D}}$.

$X=\left[x_{1},x_{2},\;\ldots ,x_{D}\right]$ being a vector of counted features (taxa) in 1 sample and *G*(*x*) the geometric mean of x. Count zero values in the initial data frame were imputed through the Geometric Bayesian Multiplicative procedure, using the zCompositions R package cmultRepl function, so that logarithms could be computed.

The sequence data have been deposited in the European Nucleotide Archive database under the accession number PRJEB63387.

### 2.4. Calculations and Statistical Analysis

Milk fat, protein, and lactose concentrations were calculated as the average of daily milking data. Energy-corrected milk (ECM) was calculated as ECM=0.25∗MY+12.2∗F+7.7∗P, where *MY* is milk production (kg), *F* is milk fat (kg), and *P* is milk protein (kg).

Methane concentration was calculated based on the eructation peaks [35,47], averaged per cow and week, and expressed in ppm. The eructation peaks and the mean of the maximum values of these peaks were calculated every time the cow entered in the AMS using a custom designed program that analyzed the cow traffic report of the AMS and the NDIR methane data report. Methane production was calculated using methane concentration (expressed in volume ppm) as described in [34]. Methane intensity was calculated as the daily methane production divided by the daily raw milk production.

To estimate the microbial N flux, the urinary excretion of allantoin and uric acid purine derivates (PD) was used [48]. To calculate the total excretion of allantoin, creatinine, and uric acid for each daily interval, the product of the volume of urine obtained and the concentration of metabolites was calculated. In addition, a mean daily creatinine rate (29.0 mg/kg BW per day) was considered with data from all cows in the trial. The sum of allantoin and uric acid excreted in the urine made up the total excretion of PD. From the BW of individual cows as 0.385 mmol/0.75 gross weight per day, the endogenous excretion of PD (mmol/d) was estimated. The total absorption of microbial purines and the ruminal synthesis of microbial N were described by [49].

For the statistical analysis, each dairy cow (n = 16) was considered as the experimental unit. SAS software was used for the statistical analyses [50]. Milk yield, ECM, milk fat and protein concentrations, and milk fat and protein yield were analyzed by a MIXED model for repeated measures assuming a covariance structure fitted on the basis of Schwarz’s Bayesian information model fit criterion. The statistical model included fixed effects of treatment (CTR vs. CHI), the initial record measured at week 0 (used as a covariate), breed and week, and the interaction between treatment and breed. The model included the random effects of cows within pairs. CH_4_ production, CH_4_ concentration, and CH_4_ intensity were analyzed using the previous statistical model but including the fixed effect of number of lactation and without including a covariate. VFA concentrations and purine derivatives were averaged by cow. VFA, serum metabolites, purine derivatives, and alfa-diversity variables were analyzed using the previous statistical model but without considering covariates or repeated measures. Treatment means were separated using a Tukey test.

The CLR-transformed data (at phylum, class, order, family, and genus) were used to perform the PERMANOVA analysis. Differences between centroid distances using treatment, breed, and their interactions as grouping variables were determined through PERMANOVA [51,52].

Sparse partial least squares discriminant analysis (sPLS-DA) was used to explore the possibility of grouping and classifying samples at genus level using mixOmics [53]. The sPLS-DA is a supervised machine learning approach that enabled us to discriminate genera that best characterize each experimental group. sPLS-DA analysis identified a subset of discriminant genera: for each genus, a loading value representing the discriminant power of the genus in explaining differences among the two different treatments (CTR and CHI) was obtained.

Relative abundances (RA) CLR-transformed of bacterial and eukaryote taxa at the phylum and genus level were analyzed using the MIXED procedure of SAS [50], using the statistical model previously described for VFA, serum metabolites, and purine derivatives analyses. Normality was tested using Shapiro–Wilk or Kolmogorov–Smirnov. *p*-values were adjusted by the Benjamini–Hochberg method to control the false discovery rate (FDR).

To investigate the correlations between the ruminal VFAs or CH_4_ and microbial genera, a regularized canonical correlation analysis (rCCA) was performed using the package mixOmics (v. 6.15.45) [53] in R (v4.0.5) [54]. To perform the rCCA analysis, the correlation values between the relative abundances CLR-transformed of microbial taxa (at genus level) and each ruminal VFA or CH_4_ value were computed to calculate a similarity matrix. A clustered image map was inferred using a similarity matrix obtained from the rCCA.

## 3. Results

### 3.1. VFA Profile

The effect of feeding CHI on VFA proportions can be seen in Table 1. No significant interaction between treatment and breed was found. Regarding treatment effects, we observed that feeding CHI decreased (*p* = 0.046) acetic acid (C2) proportions (63.3 vs. 64.7 mol/100 mol) and increased (*p* = 0.008) propionic acid (C3) proportions (18.5 vs. 16.6 mol/100 mol) compared to the control group. On the other hand, feeding CHI did not affect proportions of Butyric (C4; *p* = 0.376), valeric (*p* = 0.934), isobutyric (*p* = 0.827), isovaleric (*p* = 0.154), or total BCVFA (*p* = 0.321) compared to the control group. The changes in the main VFA led to a decrease (*p* = 0.013) in the C2/C3 ratio (3.44 vs. 3.92) and in the (C2 + C4)/C3 ratio (4.2 vs. 4.87; *p* = 0.011) in the rumen samples of the CHI group compared to the control group.

### 3.2. Milk Yield, Milk Composition and Methane Production

The effect of feeding CHI on milk yield and composition, and methane production can be seen in Table 2. No significant interaction between treatment and breed was found. Regarding treatment effects, we observed that CHI addition did not affect daily yields of milk (*p* = 0.562), ECM (*p* = 0.709), fat (*p* = 0.167), protein (*p* = 0.664), and lactose (*p* = 0.627) compared to the control group. Similarly, feeding CHI did not affect milk concentrations of fat (*p* = 0.080), crude protein (*p* = 0.358), and lactose (*p* = 0.640) compared to the control. Regarding methane emissions, CHI supplementation did not significantly affect CH_4_ concentration (*p* = 0.779), CH_4_ production (*p* = 0.492) or CH_4_ intensity (*p* = 0.616) compared to the control group.

### 3.3. Blood Parameters and N Flux

The effect of feeding CHI on blood parameters and N flux can be seen in Table 3. Except for glucose, no significant interaction between treatment and breed was found for the rest of the variables. Regarding glucose concentration, while feeding CHI tended to increase (*p* = 0.076) blood glucose concentration in HF cows (3.70 vs. 4.49 mmol/L) compared to the control group, blood glucose concentration was not different (*p* = 0.207) in BS cows of both experimental groups. Regarding treatment effect, as it can be seen, feeding CHI did not affect BUN (*p* = 0.960) or IGF-1 (*p* = 0.115) concentrations compared to the control group. Supplementing with CHI affected neither purine derivatives excretion (*p* = 0.496) or the nitrogen flux (*p* = 0.432) compared to the control group.

### 3.4. Ruminal Microbial Community

Figure 1 represents the bacterial community composition at family level in the rumen of cows when fed the two dietary treatments. Within prokaryotes, the three most abundant phyla were Bacteroidetes (64%), Firmicutes (18%), and Proteobacteria (2.6%), and, within eukaryotes, the main phyla were Ciliophora (6%) and Chytridiomycota (0.89%). The predominant family of Bacteroidetes was Prevotellaceae (39%). Within Firmicutes, the dominant families in order of importance were undefined families within the order Eubacteriales (3.6%), undefined families within Clostridia clade (3.6%), Lachnospiraceae (2.5%), and Oscillospiraceae (2.5%), whereas the Proteobacteria mainly consisted of Succinivibrionaceae (1.4%). Regarding Cliliophora, the main families in order of importance were undefined families within Ciliophora (1.5%), Stentoridae (0.68%), and Ophryoscolecidae (0.60%). The predominant family of Chytridiomycota was Neocallimastigaceae (0.9%).

The experimental concentrate with CHI did not influence microbial species richness, as expressed by different diversity indices, such as Chao1 or Shannon (Table 4). The statistical test performed with PERMANOVA at the genus level revealed no differences in microbial community between experimental concentrates, but a significant interaction between treatment and breed was observed (*p* = 0.05).

The sPLS-DA analysis did not allow us to observe a clear class prediction and group separation at the genus level (Figure 2A) since a slight match of the 95% confidence ellipses was observed. Component 1 was found to characterize rumen microbiome of cows fed the control concentrate, including *Trypanosoma*, *Methanosphaera*, *Neocallimastix*, *Piromyces*, and *Epidinium* (Figure 2B). Component 2 characterized rumen microbiome of cows fed the concentrate with CHI, identifying *Anaeroplasma*, *Coprococcus*, undefined genera within the Erysipelotrichaceae and Tannerellaceae families, *Roseburia*, *Sharpea*, *Bifidobacterium*, *Alistipes*, *Treponema*, and *Ruminococcus*, between others, as most important contributors to the group separation.

Among the different microbial phyla (Appendix A), no significant interaction between treatment and breed was observed for different phyla. A tendency (Adjp = 0.100) to increase Spirochaetes with CHI was observed.

At the genus level (Appendix A), no significant interaction between treatment and breed was observed. Among Archaea genera, CHI tended to reduce *Methanosphaera* (Adjp = 0.100) compared to CTR. Very slight changes were observed among Bacteria genera with CHI. Chitosan only increased RA of *Anaeroplasma* (Adjp = 0.046), compared to CTR.

Regarding Eukaryota genera, no significant interaction between treatment and breed was observed. CHI did not significantly affect abundances of Eukaryota genera, only tendencies to decrease *Neocallimastix* (Adjp = 0.133), *Pyromyces* (Adjp = 0.133), *Anaeromyces* (Adjp = 0.133), *Stentor* (Adjp = 0.133), *Entodinium* (Adjp = 0.133), *Epidinium* (Adjp = 0.133), *Ichthyophthirius* (Adjp = 0.133), and *Halteria* (Adjp = 0.133) were observed .

The correlations between ruminal VFA, methane concentration, and microbial taxa were represented by a Clustered image map (Figure 3) inferred from the rCCA analysis. In general, methane concentration presented weaker correlations with microbial taxa than VFA.

Genera *Ruminoclostridium*, *Methanosphaera*, *Eudiplodinoum*, and *Ruminococcus* were positively correlated with methane concentration. Genera within rumen ciliates (*Stentor*, *Polyplastron*, *Halteria*, *Ichthyophthirius*, *Isotricha*, *Entodinium*, and *Epidinium*), fungi (*Anaeromyces*, *Piromyces*, and *Neocallimastix*), and archaea (*Methanosphaera*, *Methanobrevibacter*, and undefined genera within Methanobacteriales, Methanobacteriaceae, and Methanobacteria) were positively related to acetic acid concentrartion and C2/C3 ratio in the rumen. Bacterial genera *Anaeroplasma*, *Coprococcus*, *Clostridium*, *Sharpea*, *Bifidobacterium*, *Roseburia*, *Ruminococcus*, *Leuconostoc*, *Blautia*, *Stomatobaculum*, *Alistipes*, *Butyrivibrio*, *Succiniclasticum*, and *Desulfovobrio* were positively related to propionic acid concentrations in the rumen. Genera *Ruminoclostridium*, *Methanosphaera Prevotella*, *Alloprevotella*, *Eubacterium*, *Bacteroides*, *Fibrobacter*, *Pseudobutyrivibrio*, and *Acetobacter* were positively related to butyric acid concentrations. Finally, the genera *Coprococcus*, *Bifidobacterium*, *Streptococcus*, *Pseudomonas*, *Alistipes*, and *Butyrivibrio* were positively correlated to BCVFA concentrations.

## 4. Discussion

In recent years, multiple efforts have been placed in finding novel nutritional strategies capable of reducing both enteric CH_4_ emissions and the energy losses that CH_4_ production entails during feed fermentation in the rumen, thereby improving productive efficiency. As part of the existing strategies, various nutritional alternatives have been tested with varying degrees of success. The use of feed additive chemicals, antibiotics, methane inhibitors, and plant extracts can improve animal performance [55,56,57,58]. However, the use of chemical products and the residues in animals are a source of concern due to their role in the production of human-edible products. The development of microbial resistance to antibiotics, excessive toxicity, and the cost of some products have been crucial factors that limit their use to solve the problem in question [59,60]. Therefore, new additives that can positively modify the rumen environment without negatively affecting animal health and production together with product or environmental safety are required.

In this context, the use of CHI as a feed additive has been studied in both in vitro and in vivo trials. Chitosan is a natural, non-toxic, and biodegradable biopolymer with antimicrobial properties that enables the manipulation of ruminal ecosystems [25,61,62,63,64]. The main hypothesis of its mode of action is that, due to its polycationic nature, it interacts with the negatively charged external membrane of numerous microorganisms with the positive charges of the protonated amino groups (NH^3+^), causing alterations in the cell surface, leading to leakage of intracellular substances, and ultimately resulting in cell death [65]. In this context, Gram-positive bacteria, which have a more accessible outer layer of peptidoglycan than Gram-negative bacteria, seemed to be more extensively affected by its antimicrobial action [61].

In this study, positive effects of CHI supplementation were observed regarding the profile of VFA towards energetically more favorable routes. The results of the present trial showed how the supplementation with 135 mg/BW of CHI increased the production of propionic acid, as observed by De Paiva et al. [24] in dairy cows, Dias et al. [23] in beef steers, and Goiri et al. [22] in sheep. In turn, the proportion of acetic acid decreased, as observed by Araujo et al. [66], Vendramini et al. [25], and Zanferari et al. [29], with its consequent reduction in the acetic–propionic and acetic/butyric–propionic ratio. This shift in the VFA proportions towards a more favorable fermentation routes with increased propionic acid and decreased acetic acid is one of the most reliable and repeatable results observed in the literature both in vitro and in vivo when CHI is supplemented [67,68]. According to previous results, this may occur because CHI exerts a more pronounced antimicrobial effect towards Gram-positive bacteria (cellulolytic and hemicellulolytic bacteria), with amylolytic bacteria prevailing [20,26,29].

In the current trial, a specific effect on cellulolytic bacteria was not observed. However, chitosan increased the RA of *Anaeroplasma*. It has been noted [69] that a ruminal *Anaeroplasma* sp. inhibited cellulolysis by the ruminal fungus *Neocalliinastix frontalis* by 55%. Also, Joblin and Naylor [70] reported that ruminal mycoplasmas inhibited cellulolysis by *Piromyces cominunis* and cellulolysis by *Ruminococcus albus*. Moreover, a recent study has pointed out that, despite its low relative abundance in the rumen, rumen fungi and ciliates contributed an unexpectedly large share of transcripts for enzymes capable of degrading cellulose and hemicellulose, respectively [71]. In the present study, a tendency to decrease RA of cellulolytic protozoa and fungi was observed. Other authors also have observed that CHI promoted a strong decrease in the protozoal activity in vitro [72], and, although, to the best of our knowledge, no report about the effect of CHI on rumen fungi was published, CHI has been shown to have fungicidal effects in many studies [73,74,75]. Although these effects were not statistically significant when the *p*-value was adjusted, they gave us a clue of what can be the mode of action of CHI in the rumen, which should be confirmed in future works with higher numbers of animals in order to be able to detect subtle differences between treatments.

Correlations between rumen VFA and microbial taxa, represented by a clustered image map, showed that rumen ciliates and fungi were positively correlated to acetic acid proportions and acetic to propionic acid ratio. On the contrary, *Anaeroplasma* was positively correlated to propionic acid proportions, indicating effects of these microorganisms on the shifts observed in the fermentation pattern.

Moreover, Belanche et al. [26] reported that CHI could be partially degraded due to different microbial enzymatic actions in the rumen. In fact, these authors observed an increased amylase activity when CHI was supplemented. These authors also hypothesized that the resultant chitooligosaccharides could further be used by some bacteria as a carbon source, explaining, to some extent, the greater production of propionate. In the present study, enzymatic activities have not been measured, but this hypothesis could not be ignored.

Although CHI did not influence bacterial species richness or microbial structure, the sPLS-DA revealed that rumen microbiome of cows fed the concentrate with CHI was characterized by taxa such as *Anaeroplasma*, *Coprococcus*, undefined genera within Erysipelotrichaceae and Tannerellaceae families, *Roseburia*, *Sharpea*, *Bifidobacterium*, *Alistipes*, *Treponema*, and *Ruminococcus*, with *Anaeroplasma* being the only taxa that significantly increased with CHI. On the other hand, rumen microbiomes of cows fed the CTR concentrate were characterized by genus such as *Trypanosoma*, *Methanosphaera*, *Neocallimastix*, *Piromyces*, and *Epidinium*, which were found to have a tendency to increase in the CTR group compared to the CHI-supplemented group.

There are very few studies in the literature accounting for the effects of CHI on ruminal microorganisms, and most of them are performed on in vitro conditions. This fact makes it difficult to properly compare our results with those previously reported in the literature. Moreover, to our knowledge, there is no study published regarding the effect of CHI on a whole metagenome in productive animals, providing information of the abundance of the different microbial kingdom (Eukaryote, Prokaryote, Archaea) at the same time.

It would be expected that the observed changes in the VFA or in the microbiome could have a concomitant effect on CH_4_ production. While in acetic and butyric acid production, H_2_ is released, leaving more H_2_ for CH_4_ production, in propionic production, free H_2_ is captured, leading to better energy use [3,76,77]. However, contrary to previous results reported by several in vitro studies, in which substantial reductions in CH_4_ emissions were observed [26,30], in the present study, this improvement in the fermentation pathways did not lead to a significant reduction in CH_4_ concentration, yield, or emission intensity. Although there is no statistically significant difference between the groups, the difference in methane emissions was more than 10%, maybe reflecting that there was not enough statistical power in this study to support the hypothesis. The lack of agreement between in vivo and in vitro studies may be, in part, due to the fact that the in vitro experiments did not consider the complex process of ruminal fermentation, rumen pH, the pattern of volatile fatty acids, absorption of VFA, passage rate, rumen dilution, anatomical differences of the rumen, and alterations in microbiome structure [32]. The number of experiments to quantify methane emissions when CHI is added to a ration in vivo are limited, but results in the current trial are consistent with the lack of decreased CH_4_ production reported by Henry et al. [31] in beef cattle and Jiménez-Ocampo et al. [78] in crossbred heifers.

Ciliate protozoa and rumen fungus have been found to be the main rumen microbes associated with methane emissions in dairy cattle [79,80]. Relationships between ruminal archaea and methane emissions are more contradictory, with some authors finding a positive relationship [81] and others not [79].

In the present study, although tendencies to decrease RA of some archaea such as Methanosphaera, and to decrease rumen ciliates and fungus, were observed with CHI, and, although some of these taxa were positively related to enteric methane emissions in the clustered image map, these shifts seemed not to have a significant effect on final methane emissions.

Regarding microbial N flux, our results disagreed with those reported by Gandra et al. [82], who found a decrease in microbial protein synthesis when CHI was used as additive. This could be due to differences in the CHI dose, animals, and diets since the work of [83] was carried out in beef heifers fed a concentrate diet. As Gandra et al. [82] mentioned in their work, high-concentrate diets entail a decrease in ruminal pH that may affect the efficiency of microbial protein synthesis. Moreover, the antimicrobial action of CHI was enhanced at low pH values, which potentially exerted a more pronounced effect on microbial protein synthesis. However, the obtained results agreed with those of De Paiva et al. [24], Del Valle et al. [28], and Seankamsorn et al. [64], who did not observe an effect of CHI supplementation on purine derivatives or microbial synthesis with lactating dairy cows. This result indicated that CHI supplementation had no adverse effects on microbial protein synthesis for lactating dairy cows in the conditions of this study.

Few reports are available on the effect of supplementation of CHI in the diet on milk performance of dairy cows. For instance, Zanferari et al. [29] reported a decrease in milk production when cows were supplemented with CHI and a diet rich in unsaturated fat but observed no effects on milk performance when CHI was added to a diet without unsaturated fats. On the contrary, Zheng et al. [83] reported that CHI supplementation linearly increased milk production and fat-corrected milk. These authors attributed the enhanced production to an observed higher intake. Unfortunately, we have not measured intake in the present work to corroborate this hypothesis. The lack of effect observed in the current trial agreed with Del Valle et al. [28] and Seankamsorn et al. [64], who also observed that CHI supplementation did not influence milk yield or its composition. Variation in CHI physicochemical properties, such as deacetylation degree and molecular weight [84], doses [30], rations [30] and combination with other ingredients in the diet as unsaturated fats or supplementation length in each study, may explain the observed inconsistent results. In the current trial, CHI had a 95% deacetylation degree and was dosed at 135 mg/kg BW. Other authors [28,29,85] used commercial CHI with 87% deacetylation degree but dosed CHI at 50–150 mg/kg BW. Mingoti et al. [85] used commercial CHI with 87% deacetylation degree dosed at 500–2000 mg/kg DMI. Seankamsorn et al. [64] instead used an extract-based CHI with a 98% deacetylation degree and a commercial CHI with a 90% deacetylation degree, both dosed at 2% DM intake (651 mg/kg BW). Pereira et al. [86], however, used CHI as an additive with an over 85% deacetylation degree dosed at 136–272 mg CHI/kg BW.

A deficiency of any nutrient may decrease milk production by dairy cows, but the two nutritional factors that are most likely to be limiting are energy and protein. Although it could be hypothesized that CHI would improve the productive performance of cows considering its positive effect on ruminal fermentation by increasing propionic by 11.2% and reducing the acetic–propionic ratio by 12.2%, in our study, it was observed that CHI did not reduce the energy loss associated with CH_4_ emissions, which could have been otherwise available to increase milk production. In addition, CHI supplementation did not increase microbial N flow, and protein is a crucial nutritional factor to regulate hepatic IGF-1 expression and secretion [87]. The anabolic role of plasma IGF-1 stimulating the uptake of amino acids and glucose by the cells resulting in a stimulation of milk production [88] is well established, which could explain the lack of effect of CHI supplementation on milk yield and quality. Therefore, although CHI supplementation improved the fermentation efficiency, this shift was not large enough to have a concomitant improvement in animal performance.

Although no interactions between the treatment and the breed were found for most of the studied variables, the use of two different breeds in the present study could have been a confounding factor that should be considered for future works. Moreover, studies with a higher number of animals should be necessary in order to be able to appreciate subtle differences in rumen microbial taxa that could corroborate the findings observed in the present study. Finally, as CHI effects on ruminal VFA did not have an effect on methane production and milk performance, future works with higher CHI doses would be necessary to assess the potential use of this additive as methane inhibitor.

## 5. Conclusions

In conclusion, supplementation with 135 mg CHI/kg BW did not affect rumen microbial diversity, but induced shifts in the relative abundance of some microbial taxa. Chitosan did not decrease fibrolitic bacteria but increased RA of *Anaeroplasma*, which is known to inhibit cellulolysis by some ruminal fungus and bacteria. Chitosan also tended to reduce fibrolytic fungi and protozoa, resulting in a shift of ruminal fermentation towards a lower acetic to propionic ratio. Although some archaea such as Methanosphaera tended to decrease with CHI inclusion and fermentation routes were shifted to propionate production, CH_4_ emissions, microbial protein synthesis, and milk yield or composition were not affected by CHI supplementation. The dose of chitosan used in this study seemed to induce subtle changes in microbial taxa and VFA in the rumen but seemed not to be large enough to have a concomitant improvement in animal performance and CH_4_ reduction. Therefore, future works with higher doses would be necessary to assess the potential use of this additive as methane inhibitor.

## Figures and Tables

**Figure 1 animals-13-02861-f001:**
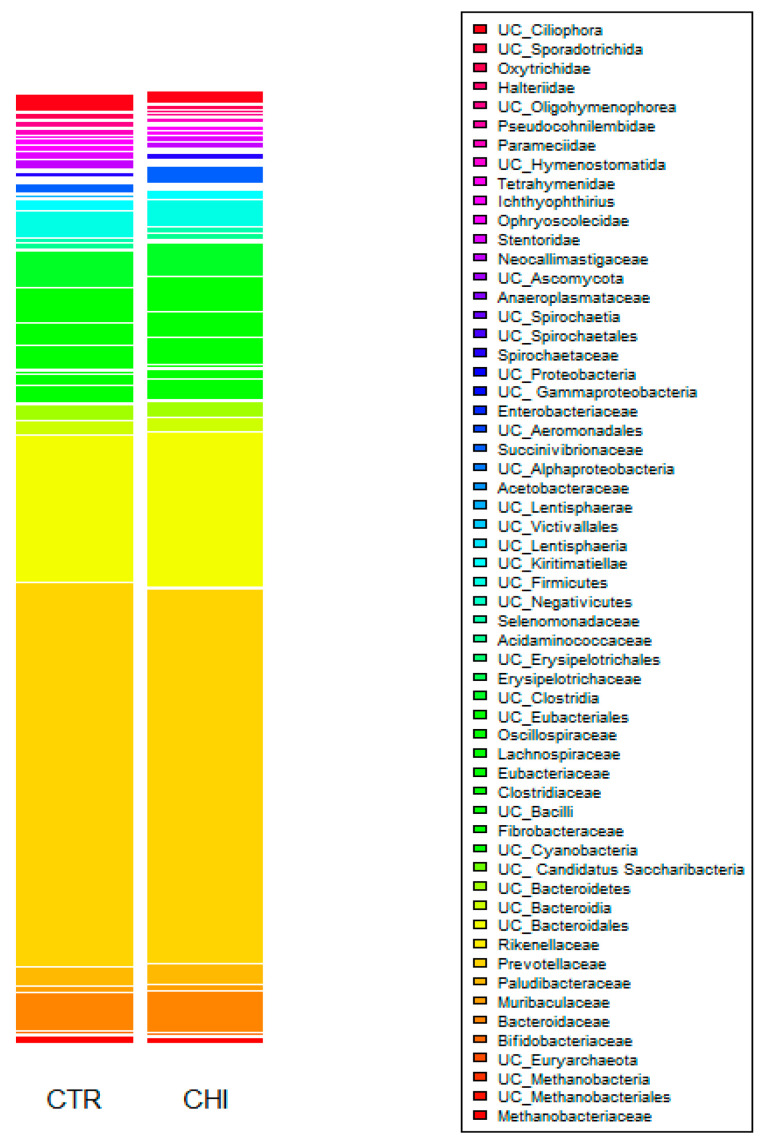
Microbial community composition at family level in the rumen of cows (n = 16) in the experimental groups: CTR (control) and CHI (chitosan).

**Figure 2 animals-13-02861-f002:**
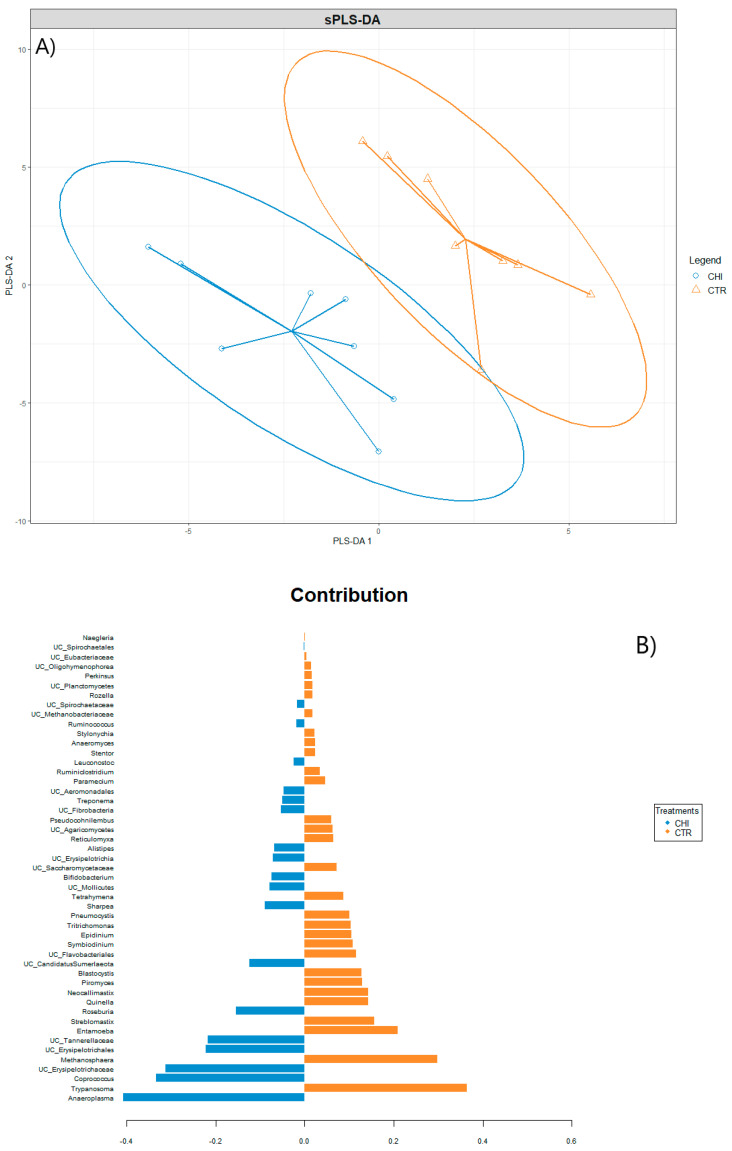
(**A**) Sample plots from Sparse Partial Least Squares Discriminant Analysis (SPLS-DA) performed on the data at genus level, including 95% confidence ellipses. Samples are projected into the space spanned by the first two components. Samples are colored by their experimental group, CTR (control) or CHI (chitosan), and (**B**) shows the features selected by the SPLS-DA algorithm as most predictive or discriminative features in the data to classify the samples and represent the contribution of each feature selected on the first (30) and second component (19), respectively, with contribution ranked from bottom (important) to top. Colors in the contribution plot indicate the experimental group with the highest median for each selected feature labelled at the genus level. The negative (resp. positive) sign on the *x*-axis represents the regression coefficient weight of each feature in the linear combination of the sPLS-DA component.

**Figure 3 animals-13-02861-f003:**
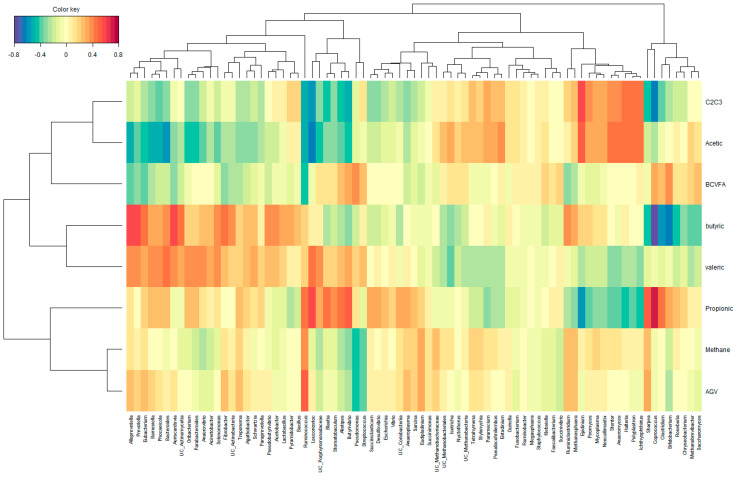
Relationships between clusters of microbial genus and ruminal volatile fatty acids and enteric methane concentration in eructated air, independently of experimental group. This clustered image map was based on the regularized canonical correlations between relative microbial abundances after CLR normalization and relative concentrations of rumen short chain fatty acids and methane concentration data. Significant correlations are colored following the key shown.

**Table 1 animals-13-02861-t001:** Effect of supplementing chitosan on ruminal volatile fatty acid proportions.

	HF			BS			*p*-Value	
	CTR	CHI	CTR	CHI	SEM	Breed	Treat	Breed × Treat
Total VFA, mM	62.9	66.6	69.8	68.4	5.04	0.418	0.823	0.628
Individual VFA, mol/100 mol								
Acetic	65.0	62.5	64.3	64.1	0.60	0.444	0.046	0.109
Propionic	16.7	18.8	16.6	18.2	0.57	0.533	0.008	0.674
Butyric	14.2	14.3	15.2	14.1	0.52	0.493	0.376	0.288
Isobutyric	0.956	0.832	0.872	0.784	0.0775	0.430	0.210	0.827
Valeric	1.28	1.25	1.29	1.26	0.054	0.877	0.635	0.934
Isovaleric	1.87	2.25	1.69	1.56	0.161	0.022	0.477	0.154
BCVFA	2.83	3.08	2.56	2.34	0.220	0.046	0.945	0.321
C2/C3	3.93	3.34	3.92	3.54	0.159	0.569	0.013	0.551
C2 + C4/C3	4.84	4.15	4.89	4.36	0.195	0.522	0.011	0.707

HF: Holstein Friesian; BS: brown Swiss; CTR: control; CHI: chitosan; SEM: standard error of the mean; Treat: treatment; VFA: volatile fatty acids; BCVFA: branched-chain volatile fatty acids; C2/C3: acetic to propionic ratio; and C2 + C4/C3: acetic plus butyric to propionic ratio.

**Table 2 animals-13-02861-t002:** Effect of supplementing chitosan on milk yield and composition, and methane production.

	HF		BS				*p*-Value	
	CTR	CHI	CTR	CHI	SEM	Breed	Treat	Breed × Treat
Yield, kg/d								
Milk	25.8	26.2	26.0	27.3	1.52	0.718	0.562	0.756
ECM	26.7	24.0	28.3	29.3	2.25	0.160	0.709	0.406
Fat	1.22	0.757	1.13	1.16	0.1536	0.355	0.167	0.131
Protein	1.05	0.912	0.944	0.993	0.1048	0.921	0.664	0.365
Lactose	1.49	1.32	1.29	1.34	0.1233	0.539	0.627	0.358
Composition, %								
Fat	4.68	3.61	4.09	4.00	0.306	0.742	0.080	0.130
Protein	3.54	3.54	3.65	3.62	0.094	0.882	0.358	0.883
Lactose	4.83	4.73	4.73	4.90	0.069	0.588	0.640	0.056
CH_4_								
ppm	1605	1770	2325	1924	329.2	0.271	0.727	0.461
g/d	214	206	290	234	45.3	0.337	0.498	0.642
g/kg milk	9.54	9.14	10.85	9.51	2.242	0.740	0.694	0.496

HF: Holstein Friesian; BS: brown Swiss; CTR: control; CHI: chitosan; SEM: standard error of the mean; Treat: treatment, and ECM: energy corrected milk.

**Table 3 animals-13-02861-t003:** Effect of supplementing chitosan on blood parameters and microbial N flux.

	HF		BS				*p*-Value	
	CTR	CHI	CTR	CHI	SEM	Breed	Treat	Breed × Treat
Blood parameters								
BUN, mmol/L	2.52	2.39	2.71	2.81	0.249	0.267	0.960	0.672
Glucose, mmol/L	3.70	4.49	3.63	3.28	0.161	0.002	0.216	0.005
IGF-1, ng/mL	141	188	103	99	12.0	<0.001	0.115	0.068
N flux								
Purine derivative excretion (mmol/d)	332	279	300	312	28.1	0.981	0.496	0.291
N flux (g N/d)	244	204	236	231	26.4	0.735	0.432	0.535

HF: Holstein Friesian; BS: brown Swiss; CTR: control; CHI: chitosan; BUN: blood urea nitrogen; SEM: standard error of the mean; and Treat: treatment.

**Table 4 animals-13-02861-t004:** Rumen microbial community diversity analysis of cows fed a concentrate with chitosan or a control concentrate.

	HF		BS			p-Value		
	CTR	CHI	CTR	CHI	SEM	Breed	Treat	Breed × Treat
Observed	681	636	652	699	63.6	0.743	0.984	0.397
Chao1	773	730	757	781	28.9	0.460	0.699	0.181
Shannon	3.01	2.59	2.59	2.75	0.180	0.405	0.387	0.077
InvSimpson	6.85	4.91	5.16	5.77	0.956	0.606	0.411	0.129

HF: Holstein Friesian; BS: brown Swiss; CTR: control; CHI: chitosan; SEM: standard error of the mean; and Treat: treatment.

## Data Availability

The data used in the study are available on request from the corresponding author.

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
