# Peer review of "Effect of Chitosan on Ruminal Fermentation and Microbial Communities, Methane Emissions, and Productive Performance of Dairy Cattle"

_animals, 2023, doi:10.3390/ani13182861_

Round 1
Reviewer 1 Report
Comments and suggestions to the authors are attached.

The manuscript would benefit from proofreading to address grammatical errors and improve the overall clarity of the writing. Some sentences are overly complex and could be simplified for better comprehension.
Reviewer 2 Report
Although the topic: methane emissions in ruminants is very important and meaningful. However, it contained very limited information in this text.
In the abstract, there are few information about the microbial communities, only description broadly without any data.
In the Introduction, some sentences are incomplete, such as line 54. It is necessary to illustrate “The formation of CH4 is a two-stage process.” in the rumen or ruminants.
Some sentences need to be supported by literature, such as line 62-63.
I disagree with your viewpoint in line 68-70. “there is a gap of knowledge regarding the identification and control of the interactions between different microbes in the rumen (i.e: Archaea, Prokaryote and Eukaryote) that will result in optimum rumen fermentation.” There are a series of studies regarding the identification and control of the interactions between different microbes in the rumen.
In line 96, [26] is irrelevant to this article, please check it.
Belanche, A.; Jones, E.; Parveen, I.; Newbold, C. A metagenomics approach to evaluate the dietary supplementation with 703 Ascophyllum nodosum or Laminaria digitata on rumen function in Rusitec fermenters. Front Microbiol. 2016, 10, 1-14. 704
On the other hand, there are a lot of mistakes with regard to the representation of references, such as “[26] and [32] under in vitro conditions reported”
In the Materials and Methods, In line 281, the authors miss a very important information about the accession number.
In the results, it is better to imporve the quality of the Figures.
In the Discussion, although the authors discussed a lot of contents, less important, significance, or novelty information contained in this text.
Overall, CH4 emissions, microbial protein synthesis and milk yield or composition were not affected by CHI supplementation, thus it is better to alter the title of this text, and emphatically discuss “Microbial Communities”.
Reviewer 3 Report
The purpose of the experiment described is not clear. I can see that 2 breeds of lactating dairy cows have been given diets with and without chitosan, but I do not understand why. The introduction has no flow from the broad context to the purpose for the experiment, and the discussion is simply some points of interest with no story.
The manuscript struggles to show what the completed work was about. I feel that there are several small stories happening simultaneously that could be part of one large cohesive story but they aren’t. It is almost as if the different facets of the experiment were done separately then collated in the manuscript rather than being created as a single project.
Another concern is the design of the experiment, or at least the description of the design. Two factors are in the design: cow breed (Holstein, Brown Swiss) and diet (control, chitosan). This indicates a 2 x 2 factorial experiment. But the number of Holstein cows is less than the number of Brown Swiss cows, so the design is unbalanced yet this isn’t acknowledged anywhere.
General points
‘Content’ is used in many instances when ‘concentration’ is meant.
There are several instances of citation strings that don’t conform with the journal style. For example [10,11,12] should be [10-12].
References – there are many instances of ‘[xx] reported something relevant’. I suggest you use ‘Author et al. [xx] reported something relevant’
Section 3.4 looks to be written by a different author to the preceding sections. US English predominates in section 3.4, but UK English predominates in previous sections.
Simple summary
At present the simple summary reads like a short version of the abstract. I suggest it be revised to be more focused to a lay audience.
Abstract
“more energetically efficient ruminal fermentation routes” assumes I know what those routes are and why they are important. Why not just report the change in VFA here?
Check conclusion – fermentation changed to more energetically favourable routes, but there was no change in anything else. If fermentation is more favourable, where is the benefit?
Introduction
The introduction is a collection of mostly unrelated paragraphs. We have methane in the context of GHG, a poor explanation of the formation of methane enterically, a brief list of mitigation strategies, the problems inherent in characterising ruminal microbes, chitosan and some benefits, then a confused link between chitosan and possible effects on the ruminal microbiome. There is no logical sequence of argument that leads me to the need for the work presented.
Also, I would like to see a hypothesis. There appears to be enough evidence for a hypothesis to be made.
L47 “In this line…” what does this mean? The previous sentence is about methane, but the first part of this sentence looks to be talking about world GHG emissions. I fail to see the connection between sentences here.
L49 Methane is not closely related to energy losses from ruminants, it is one of the pathways of energy loss.
L54 This paragraph has poor logic. I am told that “The formation of CH4 is a two-stage process.” Then am given step 1 and step 2, but step 2 only gets me to the production of hydrogen. The next sentence is a logic step backwards into the theory of methane formation in the rumen. The final sentence is only confusing. You have just told me that archaea in the rumen produce methane, now I am told that it is only likely that microbes contribute to methane yield.
L76 PCR - this abbreviation is undefined. And needn’t be an abbreviation since it appears only once.
L95 This paragraph looks like it should be the end of the previous paragraph.
L97 “reduction in CH4 emissions (-42-43%)” If you are talking about a reduction then I suspect the numbers should be positive (42 to 43%). A negative reduction is an increase.
L100 This paragraph is unfinished. Why is the information presented important?
Material and methods
L127 you cannot use the same abbreviation for the diet (CHI) and chitosan itself (CHI). Post this point it will never be clear whether the diet or chitosan is meant.
L129 The units of viscosity are not clear. I assume mPa.s is meant.
L132 Check if INRAe is correct.
L133 According to Table S1, the forage is only maize silage and grass silage. In this case, the forage portion of the diet is maize- and grass-silage, it is not based on these.
L135 how long was the covariate period? I note that cows were given the control concentrate during the covariate period but there is no indication if the animals had adapted to this before covariate measures were made.
L136 There is no defined length of the experiment, so stating that “measurements were taken during the last 5 weeks.” Doesn’t tell me how long the cows were on treatment before measurements started
L137 what is the 2004 in the location of the company?
L145 I note that chitosan powder was mixed into a portion (5 kg) of the daily concentrate allowance by hand, yet the concentrate was delivered by the AMS in a quantity dependant on the duration since the last milk and the cows maximum daily allowance. These two activities do not appear to be compatible. Please clarify how the chitosan was delivered to the animals.
L151 “mixed ration” this is the first mention of the forage being a mixed ration. If the forage was prepared in some way then this needs to be stated
L181 What is meant by ‘liquid nitrogen vapours’, was nitrogen gas intended? And ‘preserved in liquid N2 containers’, was containers of liquid N2 meant?
L182 symbol used to indicate degrees is the wrong symbol. It is º but should be °.
L182 “Besides” is used when I think “In addition” was meant.
L183 Samples cannot be “refrigerated at 4ºC and stored frozen at −20ºC”. I suspect there were placed in the fridge for some time, then stored frozen.
L191 the type of tube needs to be specified, eg: plain, heparin, EDTA. The fate of the tubes also needs to be specified, eg: were they kept on ice until analysed?
L196 I doubt the sample was “ground through a 1-mm sieve.” I suspect the samples were ground using a grinder such that the resulting material passed through a 1-mm sieve.
L210 ‘ruminal contents’ is used when I suspect ‘ruminal fluid’ is meant, especially given that earlier it was stated that the collected material had been filtered.
L211 why is ‘one’ used here when numbers less than 10 in other locations appear as the single digit?
L225 The description here should be checked. My previous experience is that samples for the collection of serum should be incubated at room temperature before centrifuging at 25°C, and that samples for the collection of plasma are immediately centrifuged at 4°C. The details given in the manuscript suggests all samples were immediately centrifuged at 4°C
L234 Were urine samples composited by day, or only within cow? There is no day effect listed in the statistical model, and there are no day results shown.
L242 I am not familiar with the processes described in this paragraph, but I note that there is lots of equipment listed that doesn’t have a manufacturer specified.
L284 this equation is not according to the journal style
L290 ‘ad hoc’ – was ‘custom’ intended?
L296 calculation of daily urine output. Assuming that the urine captured at the sampling points is the total excreted for the day appears to be poor strategy. Unless there is evidence that the collected urine volumes are the whole day, or allow calculation of the whole day, I am reluctant to accept the method of calculating N flux.
Results
No breed results are described in the text – in particular I note that HS cows produced 20% less methane (numerically) per day than BS cows. A difference of this magnitude is biologically important even if the experiment design didn’t detect this as statistically different.
L340 This paragraph would be better if the order of the results reported was the same order that they appear in the table.
L343 no indication of which result is CTR and which CHI. Assumption is that CTR result is first.
L355 This paragraph would be better if the order of the results reported was the same order that they appear in the table.
L359 Fat (1.18 v 0.954) – the CHI result here is different to that in the table. Also, while there may be no statistical difference, I note that the yield of milk fat from the cows on the CTL diet is more than 20% greater (numerically) than the cows on the CHI diet. This should be noted somewhere
L361 Fat concentration (4.39 vs 3.81) - while there may be no statistical difference, I note that the concentration of fat in milk from cows on the CTL diet is numerically 15% greater than cows on the CHI diet. This should be noted somewhere.
L362 the methane values here do not reflect those in the table. Also, I note that CHI supplementation resulted in about 13% (numerically) reduction in methane production, and a 10% (numerically) reduction in methane intensity. I appreciate that reductions in this range are difficult to detect as statistically different, even if more cows are used and methane is measured directly. However, I think there is merit in noting the numerical reductions here and discussing them later.
L420 Figure 2 caption: there is SPLS-DA and sPLS-DA, which I presume are meant to be the same abbreviation.
L430 this is the first time a tendency is mentioned, despite there being several P values in the range 0.05 to <0.10 occurring before here.
Tables
Check journal style for within-table headings
Table footnotes are usual linked to an item in the table by the use of superscript numbers.
Figures
Figure 1 – position on page
Discussion
Par 1 and 2 read like introduction. They are not discussing any of the results obtained, nor do they present any insights from the literature. Furthermore, the first paragraph finishes with a reason to do the work just described.
L503 check “sp. Inhibited”. I suspect “sp. inhibited” was meant
My understanding of the paragraphs is as follows
Par 3: shift in VFA production
Par 4: speculation about mode of action of CHI in rumen
Par 5: Correlations between rumen VFA and microbial taxa
Par 6: Looks like it should be part of the CHI mode of action, paragraph 4
Par 7: correlation between diet and microbial taxa
Par 8: more about the correlation between diet and microbial taxa
Par 9: effect of diet on methane
Par 10: this paragraph looks to have no purpose. It isn’t linked to any of the results and presents no insights. It simply states facts from the literature.
Par 11: mini correlation between microbial taxa and methane.
Par 12: N flux
Par 13: milk. There is no story here, it’s mostly a list of facts from the literature.
Par 14: energy capture
My issue with the discussion is that there is no story, no categorical statements of what was found, and there are few insights into the implication of the results.
Line specific comments
L547 “contrary to the reported by several” – it is not clear what is meant here.
L551 “The lack of agreement between in vivo and in vitro studies is common in the literature and may be, in part, due to the” is a direct copy from [31]
L557 It is stated that there was no effect in vivo also reported by [31] and [62]. [31] reported no statistical difference, but I note their results shows a 22% reduction in methane yield when chitosan was added to the diet. I suggest that there were differences, but they were too small to detect as statistically different within the power of the experiment. No methane reported in [62] – word not found using search function in PDF reader. This suggests the wrong reference is being cited
L559 “Forage intake has a direct effect on methane emissions as stated by [74] and an effect of intake on measured methane emissions cannot be ignored.” – this is true, but since milk yield can be a proxy for intake, the methane intensity results should be valid and reflect any effect of diet if there was one.
L571 “…those of and [28] who…” – there is something missing here
L602 This is a bold claim. Very few papers have reported an improvement in animal performance (extra milk or weight gain) when methane is mitigated. I am yet to see a paper that has demonstrated that energy not lost as methane was captured in a form that could be utilised by the animal.
Conclusion
This is just a summary of results. What is the implication of the research? Energy is claimed to be captured by a change in VFA profile, yet there is no effect on methane claimed, so where did that extra energy come from? And since there is no productive benefit reported, where did the extra energy captured go?
Check English. There are many unclear expressions. Examples include
· “changes induced in the rumen microbiome concerning the supplementation”. I suspect BY the supplementation is meant.
· “…deepen into the knowledge about…”. I suspect ‘explore’ or ‘expand the knowledge about’ is intended.
There are also examples of UK English (eg: maize) and US English (eg: color).
Reviewer 4 Report
Please find attached a PDF file containing my critiques and suggestions regarding the evaluation. I have marked the specific areas of concern directly on the document.Thank you for considering my feedback.

Round 2
Reviewer 2 Report
It is necessary to rewrite the simple summary, it only contains objective and results at present, without any significance of content or novelty.
It is better to add the significance test of differences between groups or LEfSE multilevel discriminant analysis of species differences in regard to ruminal microbial community. Correspondingly, authors only discuss Anaeroplasma in the text, I believe that it must contain a large number of information of biological significance using whole metagenome sequencing.
Author Response
Thanks for your valuable comments. Please check bellow our answers.
It is necessary to rewrite the simple summary, it only contains objective and results at present, without any significance of content or novelty.
In line 19-21 we indicate the novelty of the study. Simple summary has been rewritten to establish the significance of content of the the study
It is better to add the significance test of differences between groups or LEfSE multilevel discriminant analysis of species differences in regard to ruminal microbial community. Correspondingly, authors only discuss Anaeroplasma in the text, I believe that it must contain a large number of information of biological significance using whole metagenome sequencing.
In the current study we have analyzed differences between groups regarding relative abundances (RA) CLR-transformed of bacterial and eukaryote taxa at the phylum and genus level using the MIXED procedure of SAS. Moreover, we have adjusted the P value using the false discovery rate. We consider that with this analysis and the Sparse partial least squares discriminant analysis is good enough to study the differences between treatment on ruminal microbial populations, and that LEFSE multilevel discriminant analysis does not provide a further insight on the effects of CHI on ruminal microbial community. All the details was provided in the supplementary tables. We have only discussed those taxa with significant differences or tendencies between groups. As we have mentioned before we have not considered adequate to discuss on numerical changes although we agree that they could have a practical meaning. Anyway we have included in the discussion that further studies with a greater number of animals should be necessary to clearly ascertain the effect of CHI on microbial community structure.
Reviewer 3 Report
I commend the authors for their improvements to the manuscript. the resulting product is much improved.
I appreciate the quandary about reporting numerical differences that are biologically important but are not statistically different. Are the numerical differences real or not. I leave the authors to make this call and appreciate their addressing of these small, and possibly important, differences in a new paragraph in the discussion.
There are still some minor issues that I would like to see corrected.
L66 to 67: This is still in contrast to the opening sentence of this paragraph. In lines 59 to 60 I am told that methane is formed by the action of …’bacteria, protozoa, and fungi.’ Yet the closing sentence of the paragraph states that it is only ‘likely’ that microbial communities contribute the methane production. I recommend the last sentence of this paragraph be deleted.
L138 ‘…randomly assigned to CHI, supplemented with 135 mg CHI/kg BW per day, or control group, without supplementation.’ In this context, CHI appears to be a diet name. Perhaps ‘…randomly assigned to the control concentrate only, or the control concentrate supplemented with 135 mg CHI/kg BW per day.’
L144 ‘INRAe’. Hmm, the heading on the INRAE webpage is not clear at first glance. However, subsequent uses of the abbreviation later in the page are all ‘INRAE’. I recommend changing the text to ‘INRAE’.
L209 ‘ground’ – duplicate word.
L239 Blood. This paragraph is much clearer, but could you please indicate which tubes were processed by each method.
L364 I understand the VFA ratio because I am familiar with this field but to be mathematically correct, and eliminate any misinterpretation, the expression should be (C2+C4)/C3
L495 This is the first time in vitro and in vivo have appeared in italics. Please be consistent throughout the manuscript.
L585 here and subsequently, please add the author name to [xx] references to improve the readability. For example [84], [86], [31] and others
Author Response
Thanks a lot for your valuable comments. Please check bellow our answers.
I appreciate the quandary about reporting numerical differences that are biologically important but are not statistically different. Are the numerical differences real or not. I leave the authors to make this call and appreciate their addressing of these small, and possibly important, differences in a new paragraph in the discussion.
In line 567-569 we have tried to address this point
There are still some minor issues that I would like to see corrected.
L66 to 67: This is still in contrast to the opening sentence of this paragraph. In lines 59 to 60 I am told that methane is formed by the action of …’bacteria, protozoa, and fungi.’ Yet the closing sentence of the paragraph states that it is only ‘likely’ that microbial communities contribute the methane production. I recommend the last sentence of this paragraph be deleted.
The last sentence has been deleted as suggested. This action also required the deletion of a reference. As a consequence, the reference list has been updated.
L138 ‘…randomly assigned to CHI, supplemented with 135 mg CHI/kg BW per day, or control group, without supplementation.’ In this context, CHI appears to be a diet name. Perhaps ‘…randomly assigned to the control concentrate only, or the control concentrate supplemented with 135 mg CHI/kg BW per day.’
The sentence has been reworded as suggested
L144 ‘INRAe’. Hmm, the heading on the INRAE webpage is not clear at first glance. However, subsequent uses of the abbreviation later in the page are all ‘INRAE’. I recommend changing the text to ‘INRAE’.
INRAe has been changed to INRAE
L209 ‘ground’ – duplicate word.
Deleted
L239 Blood. This paragraph is much clearer, but could you please indicate which tubes were processed by each method.
Information on which tubes were processed by each method has been added.
L364 I understand the VFA ratio because I am familiar with this field but to be mathematically correct, and eliminate any misinterpretation, the expression should be (C2+C4)/C3
Reviewer is right and we have corrected it in the new version
L495 This is the first time in vitro and in vivo have appeared in italics. Please be consistent throughout the manuscript.
Corrected
L585 here and subsequently, please add the author name to [xx] references to improve the readability. For example [84], [86], [31] and others
Reference readability has been checked in those examples and elsewhere in the discussion
Reviewer 4 Report
Thanks for the changes you have made to the revised manuscript.
Author Response
Thanks to you for your valuable comments